



# Dust sputtering within the inner heliosphere

Carsten Baumann[1,2], Margaretha Myrvang[1], and Ingrid Mann[1]

[1]UiT The Arctic University of Norway, Space Physics Group, Postboks 6050 Langnes, 9037 Tromsø
[2]Deutsches Zentrum für Luft- und Raumfahrt, Institut für Solar-Terrestrische Physik, Neustrelitz, Germany

**Correspondence:** Carsten Baumann (carsten.baumann@dlr.de)

**Abstract.** The aim of this study is to investigate how sputtering by impacting solar wind particles influences the lifetime of dust particles in the inner heliosphere near the Sun.

We consider three typical dust materials: silicate, $Fe_{0.4}Mg_{0.6}O$ and carbon and describe their sputtering yields based on atomic yields given by the Stopping and Range of Ions in Matter (SRIM) package. The influence of the solar wind is character-
ized by plasma density, solar wind speed and solar wind composition and we assume for these parameters values that are typical for fast solar wind, slow solar wind and CME conditions to calculate the sputtering lifetimes of dust. To compare the sputtering lifetimes to typical sublimation lifetimes we use temperature estimates based on Mie calculations and material vapour pressure derived with the chemical equilibrium code MAGMA. We also compare the sputtering lifetimes to the Poynting-Robertson lifetime and to the collision lifetime.

We present a set of sputtering rates and lifetimes that can be used for estimating dust destruction in the fast and slow solar wind and during CME conditions. Our results can be applied to solid particles of a few nm and larger. The sputtering lifetimes increase linearly with the size of particles. We show that sputtering rates increase during CME conditions, primarily because of the high number densities of heavy ions in the CME plasma. The shortest sputtering lifetimes we find are for silicate, followed by $Fe_{0.4}Mg_{0.6}O$ and carbon. In a comparison between sputtering and sublimation lifetimes we concentrate on the nanodust
population. The comparison shows that sublimation is the faster destruction process within 0.1 AU for $Fe_{0.4}Mg_{0.6}O$, within 0.05 AU for carbon dust and within 0.07 AU for silicate dust. The destruction by sputtering can play a role in the vicinity of the Sun. We discuss our findings in the context of recent F-corona intensity measurements onboard Parker Solar Probe.

## 1   Introduction

New observations onboard Parker Solar Probe (PSP) raised an interest again in the dust destruction in the vicinity of the Sun. The corona observations with WISPR (Howard et al., 2019) onboard PSP include the F-corona that is produced by circumsolar dust. The observed corona intensity decreases with decreasing PSP distance from the Sun and this slope changes at 17 solar radii; dust depletion is mentioned as one of the possible explanations for this observation (Howard et al., 2019). While it seems established that a dust-free zone around the Sun forms because of dust sublimation, model calculations predict that for most



materials the dust free zone would be within 15 solar radii, often even within 10 solar radii (Mann et al., 2004). For dust

destruction at larger distances, the sputtering process becomes important.

Parker Solar Probe (Fox et al., 2016) and Solar Orbiter (Müller et al., 2013) will help to quantify the dust component in

the inner heliosphere with unprecedented detail. These spacecraft do not carry dedicated dust sensors, but can measure the

dust component from the F-corona intensity as mentioned above and detect high velocity dust impacts on the satellite body

using electric field sensors. The dust impacts are observed because they change the floating potential of the spacecraft for short

periods of time (see e.g. Zaslavsky, 2015).

The FIELDS instrument (Bale et al., 2016) detects dust impacts onto the PSP spacecraft. The expected signals due to dust

impacts in the vicinity of the Sun were recently considered based on the results of previous space observations and laboratory

studies (Mann et al., 2019). Analysis of FIELDS observations during the three first perihelion passages suggests that the dust

density within about 50 solar radii varies by about 50 percent between the different encounters (Malaspina et al. personal

communication). Dust destruction by sputtering is an efficient process within 50 solar radii and because the sputtering rates

depend on the solar wind conditions, it can vary with time. In addition, Szalay et al. (2020) report the first interpretations on

the PSP dust impact data. They explain the measured impact count rate as $\mu$m sized dustflux of $\beta$-meteoroid type. Since the

signals that are observed are due to the charge production during dust impacts, there is no direct information on the size of dust

particles. Other space missions also observe nm-sized dust (e.g. Meyer-Vernet et al., 2009a, b).

Sputtering, i.e., the emission of atoms from a surface due to the impact of energetic ions, occurs within the whole heliosphere

as solar wind particles hit dust particles. Sublimation of dust, i.e. the phase transition of a body due to absorption of solar

radiation and subsequent increase of its vapour pressure, happens only when the equilibrium temperature exceeds the binding

energy of the atoms in the dust structure.

Within the heliosphere, solar radiation and energetic particles are intensive enough to form thin abrasive exospheres as

at Mercury (e.g. McGrath et al., 1986; Wurz, 2012) or moons (e.g. Wurz et al., 2007; Killen et al., 2012; Vorburger et al.,

2019). Dust in the heliosphere outside of 1 AU is more affected by sputtering, examples are the dust on the surface of comet

67P/Churyumov-Gerasimenko (Wurz et al., 2015) or ice grains in Saturn's magnetosphere (Johnson et al., 2008).

Analysis of astronomical observations point to the existence of nanometer-sized dust particles in debris disks around other

stars (e.g. Su et al., 2013). Theoretical considerations suggest that the nanodust is trapped under certain conditions in orbits

around the Sun (Czechowski and Mann, 2010; Stamm et al., 2019). While the trajectory of dust particles are influenced by the

bombardment of solar energetic particles (Ragot and Kahler, 2003), our work concentrates on the survival of nanodust during

passages of Coronal mass ejections (CME) and the solar wind. Czechowski and Kleimann (2017) carried out dust trajectory

calculations within a CME scenario and find trapped as well as ejected nanodust trajectories. However, the vast amount of

energetic plasma ejected during a CME does not leave the nanodust untouched. We investigate dust destruction by sputtering

and consider the conditions near the Sun, for which this process becomes important in comparison to the sublimation of dust

particles.

This study is organized as follows: Section 2 introduces the solar wind and CME composition as well as plasma densities

used in this study. This section also covers the sputtering process of dust within our solar system. Section 3 investigates the





sublimation process for dust approaching the vicinity of the Sun. The comparison of both dust loss processes and its implication for dust near the Sun is described within Section 4. Finally, Section 5 draws the conclusions of this study.

## 2 Dust sputtering

Sputtering is the physical process of atom ejection from a solid through the bombardment of energetic ions (Behrisch and Eckstein, 2007). This process usually is performed within a laboratory environment where a cathode is bombarded with noble

gas ions and the ejected cathode atom deposit and form high quality surfaces. However, this process is also well known in the context of dust destruction for interplanetary (e.g. Mukai and Schwehm, 1981) and interstellar dust grains (e.g. Barlow, 1978; Draine and Salpeter, 1979).

For the calculation of nanodust sputtering, we divide our study into three different sputtering scenarios. These are the slow solar wind conditions, fast solar wind conditions and CME conditions. In the following, the heliospheric conditions of these

scenarios are discussed in detail. Subsequently, we introduce the calculation of the dust's sputtering lifetimes. This is followed by an analysis of dust sputtering at 1 AU and in the inner heliosphere.

### 2.1 Heliospheric conditions

Figure 1 shows the composition of the three solar wind scenarios considered. The SW/CME composition used here is based on the work of Killen et al. (2012, their Table 6) and contains ten different species including protons (H), Helium ions (He) and

the heavier species Carbon (C), Oxygen (O), Nitrogen (N), Iron (Fe), Neon (Ne), Magnesium (Mg), Silicon (Si) and Sulfur (S) ions. Generally, the solar wind is composed of a big fraction of protons, a small fraction of Helium and traces of the heavier species. The composition changes between 95% H / 4.5% He for the slow solar wind and 98% H / 1.5% He for the fast solar wind, with heavier species around 0.5%. The composition of CME's however is much more dominated by heavier species, 66% H 30% He and 4% heavier species. In addition to the plasma composition, also the plasma speed and density is different

for each solar wind scenario. Table 1 summarizes the values that have been used for the solar wind / CME conditions within this study. It has to be noted that the composition, speed and plasma density of the solar wind or CMEs is highly variable. The given values represent average conditions.

**Table 1.** Solar wind properties as used within this study.

|         | plasma density $n_p$ | speed $v_p$ |
|---------|----------------------|-------------|
| slow SW | $8\,cm^{-3}$         | $300\,km/s$ |
| fast SW | $3\,cm^{-3}$         | $800\,km/s$ |
| CME     | $70\,cm^{-3}$        | $500\,km/s$ |

Sputtering is the impact process of an energetic ion or atom on a target and the subsequent removal of target atoms from its surface. The sputtering yield is the main parameter of the sputtering process itself. This yield denotes the number of target

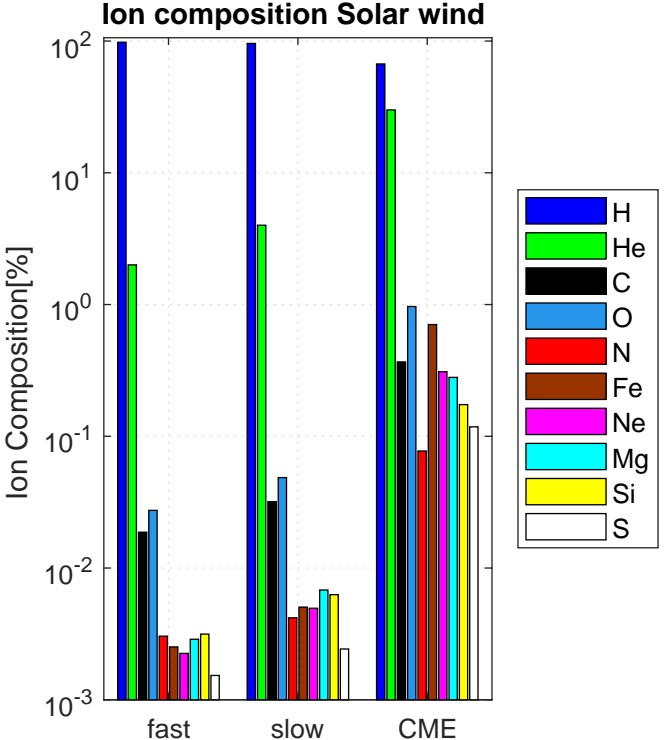

**Figure 1.** Mean composition of the fast and slow solar wind and CME's (data taken from Killen et al., 2012).

atoms sputtered by one incident ion and is a function of the ion's kinetic energy and the targets material properties (e.g. Behrisch and Eckstein, 2007, and references therein).

For further analysis we have retrieved the sputtering yield for carbon, $Fe_{0.4}Mg_{0.6}O$ and astronomical silicate ($MgFeSiO_4$) hit by nine different SW ions. For the $Fe_{0.4}Mg_{0.6}O$ and silicate we have used the SRIM (Stopping and Range of Ions in Matter) package (Ziegler et al., 2008), that derives the sputtering yields and also stopping powers and ranges of ions within compounds.

For each solar wind scenario, the SRIM program has been initialized by the above discussed plasma composition and speed (energy/nucleii). In order to derive the sputtering yield for the dust species i for a given scenario, we summed up the sputtering yields for each atom j sputtered by solar wind ion k.

$$Y_i = \sum_{j,k} Y_{i,j,k}. \tag{1}$$

The index j denotes the target atoms Mg/Fe/Si/O for astronomical silicate and Fe/Mg/O for the $Fe_{0.4}Mg_{0.6}O$ composition.

The yields correspond to the atomic ratios of the dust composition and the ion ratio of the solar wind composition. For the monoatomic carbon case we have used the analytic formula from (Eckstein and Preuss, 2003) together with the experimentally fitted sputtering parameters (Behrisch and Eckstein, 2007, and references cited within pages 45-46). The result of these calcu-

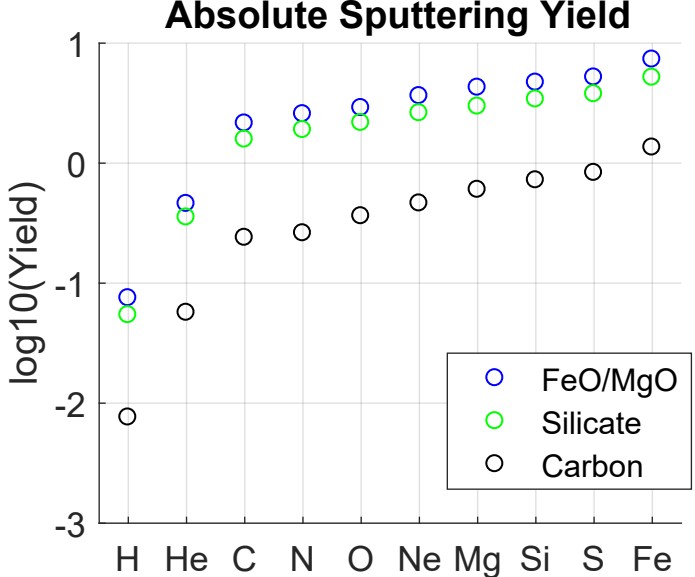

**Figure 2.** Sputtering yield for $Fe_{0.4}Mg_{0.6}O$ (blue), silicate (green) and carbon (black) as a function of solar wind ion composition and its mean speed (yields for slow SW conditions are given).

lations are sputtering yields for silicate, $Fe_{0.4}Mg_{0.6}O$ and carbon for the three different sputtering scenarios, i.e. slow SW / fast SW / CME. The individually derived yields can be found in the supplemented material.

Figure 2 shows the results of the assessment, i.e. the sputtering yield as a function of ion species (H - Sulfur S), for the three different materials. The results indicate a strong ion mass dependence as well as a target material dependence. The sputtering yields for carbon are significantly lower than for the silicate and $Fe_{0.4}Mg_{0.6}O$ materials. Furthermore, the sputtering yield for Iron ions are two orders of magnitude higher than the sputtering by light protons, that behavior is valid for all target materials. We derived the sputtering yields for all speeds stated in Tab. 1, with the 300 km/s case resulting in the highest values and

500 km/s being 20% lower and the 800 km/s 40% lower. The effect of ion impact speed on the sputtering yield is rather low compared to the importance of SW composition or impact material.

     To show the importance of heavy ion sputtering especially during CME conditions we have calculated the relative sputtering yield $Yr_k$ weighed with the plasma composition as follows:

$$Yr_k = \frac{c_k \cdot Y_k}{\sum c_k \cdot Y_k}. \qquad (2)$$

$c_k$ denotes the composition of the solar wind plasma as shown in Fig. 1. The relative sputtering for the case of carbon as target material is shown in Fig. 3, the other target materials show similar results. It is evident that the slightly higher amount of heavier ions within the CME plasma result in a much higher contribution of the sputtering yield from these ions according to Fig. 2. It can be expected that this effect will enhance the sputtering effectiveness enormously. Usually, when considering



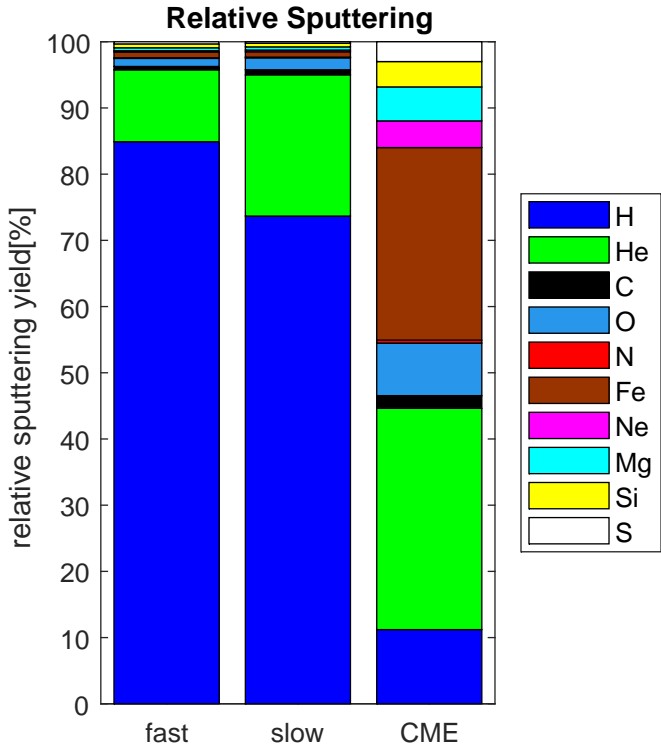

**Figure 3.** Weighed relative sputtering yield of carbon for the three different solar wind compositions, results for astronomical silicate and $Fe_{0.4}Mg_{0.6}O$ are similar, but the absolute values vary according to Fig.2.

the fast and slow SW it is reasonable to account only for H and He sputtering (e.g. Wurz et al., 2010). However, we have also

included the heavier ions in our sputtering calculations for the fast and slow SW scenarios.

The sputtering yields used in this study have only been derived by considering a normal impact of an ion onto the target particle and a resulting collision cascade governed by atomic forces within the target lattice. However, it has to be noted that the sputtering process is also heavily influenced by a number of additional parameters, which cannot be accounted for in this study. Eventual sputtering yield enhancement can occur due to high target temperatures (Roth and Möller, 1985), non-normal

impact angle, ion charge state for sputtering of insulators (Hayderer et al., 2001), and a size dependence of the yield when considering nanometer sized dust (Järvi et al., 2008). All these processes might enhance the sputtering yield substantially. On the other hand microscopic surface roughness can increase but also decrease the sputtering yield under certain circumstances, e.g. slant sputtering (Ruzic, 1990). In addition, we also do not consider composition change due to eventual implementation of solar wind ions into the nanodust or fractional depletion of a certain atom type within the dust.





## 2.2 Sputtering lifetimes

For the derivation of nanodust sputtering lifetimes we follow the formalism given by Wurz (2012). The mass loss rate from a surface through sputtering in the solar wind is given by the following:

$$\frac{dm_{sput}}{dt} = -f_{SW} Y_{tot} A m_A. \tag{3}$$

Here, A is the cross section of the dust, $f_{SW}$ the solar wind ion flux, $Y_{tot}$ the total sputtering yield of the target material and $m_A$ is the mean mass of the sputtered atoms.

Under the assumption of constant composition and size independent sputtering yield, the sputtering lifetime can be integrated from the sputtering mass loss rate of a circular surface exposed to the SW/CME plasma:

$$t_{sput} = \frac{4 r_0 N_A \rho}{f_{SW} Y_{tot} M}. \tag{4}$$

Here, $r_0$ is the initial radius of the dust, $N_A$ is the Avogadro constant, $M$ and $\rho$ are the molar mass and mass density of the sputtered material. For the solar wind flux $f_{SW} = n_p \cdot v_p$ the values from Tab. 1 have been used.

Figure 4 shows sputtering lifetimes of 1 nm dust at a distance 1 AU from the Sun. Lifetimes for all three plasma conditions and the three different dust compositions are shown.

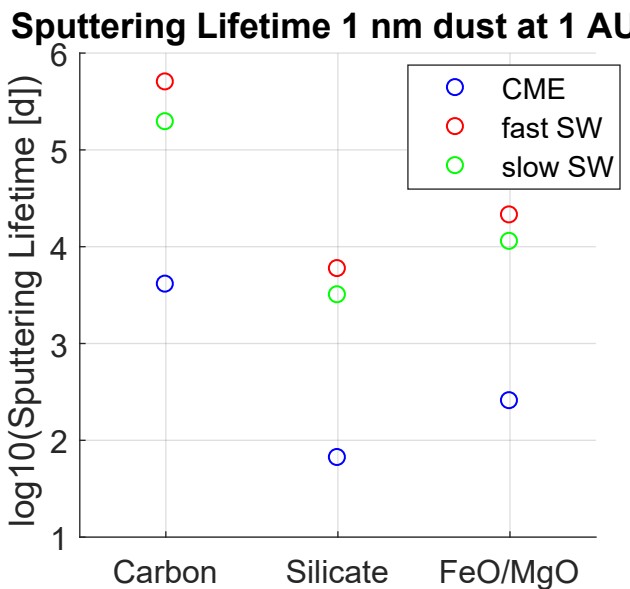

**Figure 4.** Sputtering lifetimes for carbon, silicate and $Fe_{0.4}Mg_{0.6}O$ dust particles at 1 AU and size 1 nm, line color indicate solar wind / CME composition and flux.



One can see that carbon nanodust survives longest among all three studied composition, i.e. 1 nm dust survives 5000 days at 1 AU under CME conditions. $Fe_{0.4}Mg_{0.6}O$ sputtering lifetimes are by a factor of 20 shorter. The lifetimes of silicate are

a factor of 60 shorter than the carbon sputtering lifetimes. These factors vary slightly with all solar wind condition. When comparing the different solar wind conditions, CME sputtering lifetimes are the shortest. Sputtering lifetime in the slow solar wind are 20 fold longer. The lifetimes for fast solar wind conditions are 20 times longer than the lifetimes in CME condition. This behavior varies a bit from one dust composition to the other. The short lifetimes in the CME scenario occur due to the presence of heavy ions in an overall denser plasma cloud. However, CME's are distinct solar eruptions and these sputtering

conditions do not last longer than one or two days and occur only locally in the heliosphere. The given lifetimes of several ten days and more at 1 AU for 1 nm dust make a full destruction due to CME sputtering not possible. However, for the case of fast and slow SW, which is present within the heliosphere at all times, the sputtering life times are close to ten orbital periods in the case of silicate nanodust and thirty years for $Fe_{0.4}Mg_{0.6}O$ nanodust.

The sputtering lifetime described in Eq. 4 is linear in initial dust particle radius and enables easy calculation of lifetimes for

other dust sizes. In Fig. 5 we show the derived lifetimes of dust particles in the size range from 1 nm to 1 $\mu$m at the Earth's orbit. Sub-micron dust particles at Earth's orbit have sputtering lifetimes which can reach several hundred thousands of days,

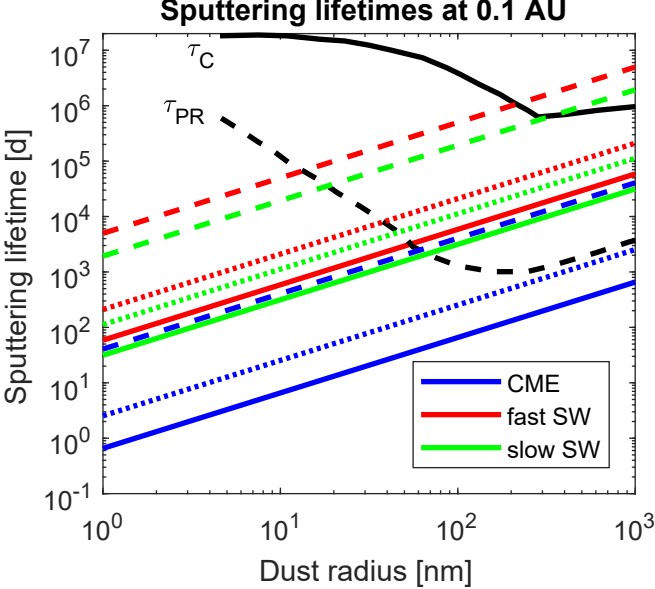

**Figure 5.** Sputtering lifetimes for carbon(dashed), silicate(solid) and $Fe_{0.4}Mg_{0.6}O$(dotted) dust particles at 1 AU for different sizes, line color indicate plasma environment (blue - CME conditions, green - slow solar wind, red - fast solar wind). Collisional lifetime ($\tau_C$) and Poynting-Robertson lifetime ($\tau_{PR}$) are shown in black for comparison (data taken from Grün et al., 1985).

i.e. thousands of orbital periods. For a better comparison, the sputtering lifetimes are plotted together with the collisional and Poynting-Robertson lifetimes given by Grün et al. (1985). As mentioned above, only nanodust in the small size limit





can be significantly removed by solar wind sputtering in reasonable timescales. For slow and fast solar wind conditions at

1 AU, we find that the sputtering lifetime of silicate particles smaller than 60 nm is clearly below their Poynting-Robertson and collision lifetime. That is also the case for $Fe_{0.4}Mg_{0.6}O$ dust below 30 nm and carbon dust below 20 nm. We point out, that for CME conditions at 1 AU, we also find that the sputtering lifetime of silicate and $Fe_{0.4}Mg_{0.6}O$ particles is well below Poynting-Robertson and collision lifetime of the dust in the whole considered size interval of 1 to 1000 nm. In practice, this has no consequence because of the short time duration of CME. This situation changes when considering sputtering at shorter

distances from the Sun, as the SW and CME plasma density increases.

For this approach we consider a SW plasma density following a power law with exponent minus two:

$$f_{SW}(d) = f_{SW}(1\,\mathrm{AU})d^{-2}. \tag{5}$$

Here, the distance from the Sun $d$ is given in astronomical units. The used exponent lies within the range of published values, e.g. Maksimovic et al. (2005) report a value of -2.2±0.1. This values was found for the fast SW conditions which we are going

to apply for the slow SW and CME conditions as well. Figure 6 shows the lifetime of 1 nm dust at distances from the Sun from 0.01 AU to 1 AU, derived for the three different SW scenarios and three dust materials. The high vulnerability of silicate to sputtering is visible here too as their solar wind sputtering lifetimes are in the range of the carbon's lifetimes for CME conditions.

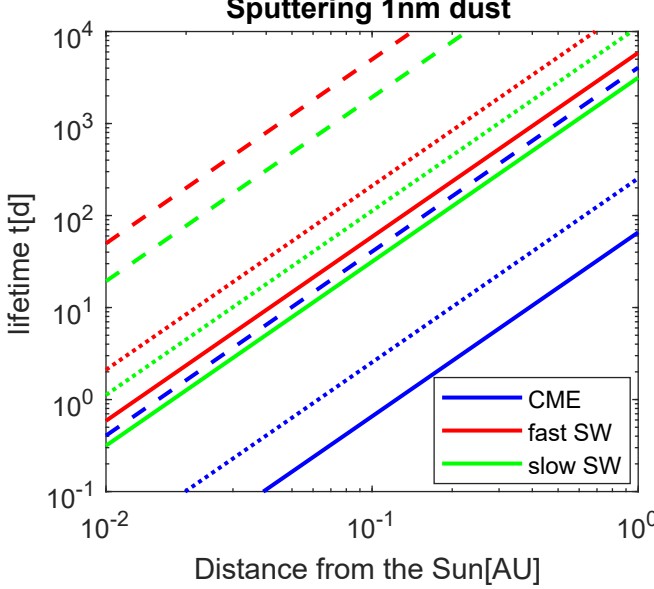

**Figure 6.** Sputtering lifetimes for 1 nm dust for the three different solar wind conditions (blue - CME conditions, red - fast solar wind, green - slow solar wind) as a function of distance from the Sun, line styles indicate the dust compositions ($Fe_{0.4}Mg_{0.6}O$ - dotted, silicate - solid, carbon - dashed)





**Table 2.** Dust erosion rate for the three compositions and solar wind conditions, rates are given for 0.1 AU, the erosion rate is independent of dust size and decreases quadratic with distance from the Sun.

| erosion rate [$\mathrm{nm\,d^{-1}}$] | $\mathrm{Fe_{0.4}Mg_{0.6}O}$ | Silicate | Carbon |
|:---:|:---:|:---:|:---:|
| fast SW conditions | $5.0 \cdot 10^{-3}$ | $1.8 \cdot 10^{-2}$ | $2.1 \cdot 10^{-4}$ |
| slow SW conditions | $9.3 \cdot 10^{-2}$ | $3.3 \cdot 10^{-2}$ | $5.4 \cdot 10^{-4}$ |
| CME conditions | $0.41$ | $1.6$ | $2.6 \cdot 10^{-2}$ |

As stated above, carbon is a very resistant material with respect to sputtering. Carbon dust with only 1 nm can survive several
ten days at 0.1 AU. Only in the case of sputtering within the CME conditions the carbon sputtering lifetimes is below the typical
duration of a CME of 1-2 days within the shortest distances from the Sun.

From the mass loss rate (Eq.3) it is also possible to derive the erosion rate of a dust particle due to sputtering. This erosion
rate, i.e. the shrinkage of dust per unit time (dr/dt), is also independent of dust size.

$$\frac{dr}{dt}(d) = -\frac{f_{SW}(d)M}{4N_A\rho} \tag{6}$$

For a distance of 0.1 AU, we derived the erosion rates of the three dust components for the three solar wind conditions in Tab.
2.

As the dust erosion rate (Eq. 6) is independent of initial dust radius, the sputtering of dust larger than 1 micron can also be
considered. For example a silicate dust particle with a size of 10 $\mu$m has a 10000 fold lifetime of a 1 nm dust grain. When
assuming the dust is in at a distance of 0.1 AU the 1 nm dust survives 0.6 days under CME conditions, i.e. it will be destroyed
by a single CME. A dust grain of 10 $\mu$m size has a lifetime of 6000 days under CME condition. That means this can be hit by
3000 strong CMEs at a distance of 0.1 AU until it will be finally destroyed. Within our solar system, CME rates vary during a
solar cycle . The rate can peak up to 400 per month during high solar activity and can be as low as 10 CME per month during
solar minimum (Lamy et al., 2019). When assuming a mean value of 100 CMEs per month, the duration a 10 $\mu$m can survive
at 0.1 AU is at least 2.5 years. It has to be noted that this requires that the dust particle is hit by every CME ejected by the Sun.
This seems to be unlikely due to the randomness of the CME propagation and its allocated size within the heliosphere. Another
reason why the lifetime of bigger dust particles might be unrealistic is that during this period the dust size and its orbit changes
drastically. This leads to a different sputtering environment and the assumption of a constant erosion rate breaks down.

As dust particles approach the vicinity of Sun their temperature increases drastically. To investigate the relevance of the
sputtering process, the seemingly low nanodust sputtering lifetimes has to be compared to dust destruction by sublimation into
free space.

## 3 Dust sublimation

The processes of sublimation, evaporation and condensation is usually described by Langmuir's equation of evaporation:





$$\frac{dm_{sub}}{dt} = -p_v \sqrt{\frac{M}{2\pi R T}} A. \tag{7}$$

In case of sublimation, it describes the sublimated mass per time unit as a function of vapour pressure $p_v$ and temperature T of

the sublimating material. $A$ is the whole surface of the dust and R is the gas constant. In the context of free space, atoms leave

the materials surface into the vacuum, while the adsorption of atoms onto surfaces can also occur under certain conditions,

e.g. the resupply of Saturn E rings by the adsorption of Enceladus water vapour (e.g. Hansen et al., 2006). The sublimation of

dust particles has been studied within different astrophysical context, e.g. protoplanetary systems (e.g. Duschl et al., 1996) and

interstellar dust (e.g. Draine and Salpeter, 1979). For a self-consistent study, the sublimation of the same dust materials as in the

sputtering part will be considered. In order to quantify the dust sublimation two parameters are needed, i.e. dust temperature

at certain distances from the Sun and the dust materials vapour pressure. For deriving the dust temperature we assume the

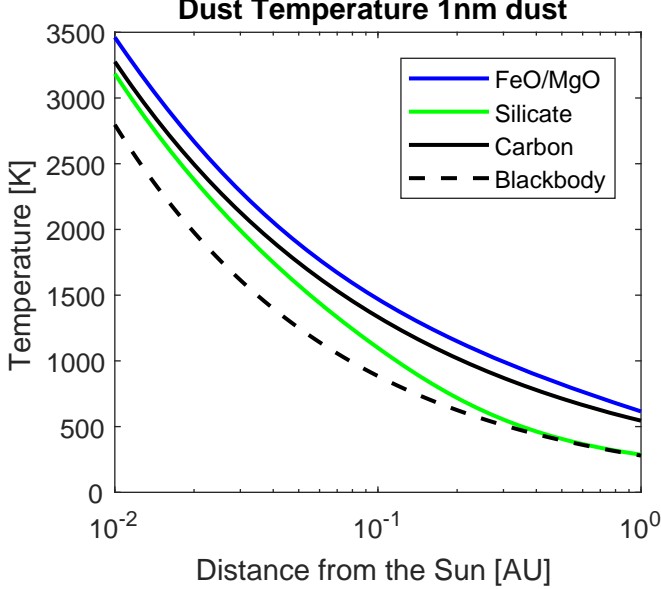

**Figure 7.** Temperature of 1 nm dust particles as a function of distance, line color indicate dust composition.

equilibrium of absorbed solar radiation and re-emitted thermal radiation of the dust particle (Myrvang, 2018). The effect of

dust cooling due to evaporation has been quantified to be only 10% of the re-emitted power (e.g. Schwehm, 1980), which we

neglect in this study. Figure 7 shows the temperature of 1 nm dust particles made of carbon, silicate and $Fe_{0.4}Mg_{0.6}O$, for

comparison the temperature of a black body is also shown. All nanodust is significantly hotter than a blackbody, except for

silicate near 1 AU which has similar equilibrium temperatures. Near the Sun, the dust temperatures of all materials exceed the

black body. At 0.01 AU the $Fe_{0.4}Mg_{0.6}O$ 1 nm dust is $\approx 700$ K hotter than a black body, carbon 500 K and silicate 400 K. All

three materials with a dust size of 1 nm are hotter than 3000 K near the Sun. The temperature change from 1 nm to 100 nm

is below 100 K for each dust material(not shown). These temperatures have been derived using Mie theory and the refractive





indices for carbon and astronomical silicate come from Li and Greenberg (1997). The refractive index for FeO/MgO is from Henning et al. (1995), we have used the data for the $Mg_{0.6}Fe_{0.4}O$ compound.

The second quantity for the description of sublimation is the vapour pressure. For the derivation of the vapour pressure for the oxides $Fe_{0.4}Mg_{0.6}O$ and astronomical silicate we used the MAGMA code (Fegley Jr and Cameron, 1987; Schaefer and Fegley, 2004). The program is very flexible with regard to material composition and the derived vapour pressures have

been checked with a vast number of experimental data. The MAGMA code has been used mainly for the change of planets and planetesimals due to geological activity but also for the evaporation of meteoroids within the Earth's atmosphere (e.g. Schult et al., 2015). The MAGMA model is a multicomponent gas-melt chemical equilibrium code and is able to derive vapour pressures for mixtures of its base components ($MgO$, $SiO_2$, $FeO$, $CaO$, $Al_2O_3$, $Na_2O$, $TiO_2$, $K_2O$, $ThO_2$, $UO_2$, $PuO_2$). The results of the MAGMA model has been successfully compared to experimental work on the vapourisation of chondrite type

material. The good performance of the MAGMA model encouraged us to use it in the context of dust sublimation near the Sun as well. The vapour pressure for carbon was used from the literature (Leider et al., 1973; Lide, 2003). Figure 8 (blue lines and left y axis) shows the vapour pressure of all three materials in the temperature range between 500 and 3000 K. The exponential growth of the vapour pressure with temperature is a typical behaviour of all materials. Please pay attention to the comparably low vapour pressure of carbon compared to the oxides, this will have an impact on the dust lifetime.

**Sublimation lifetime**

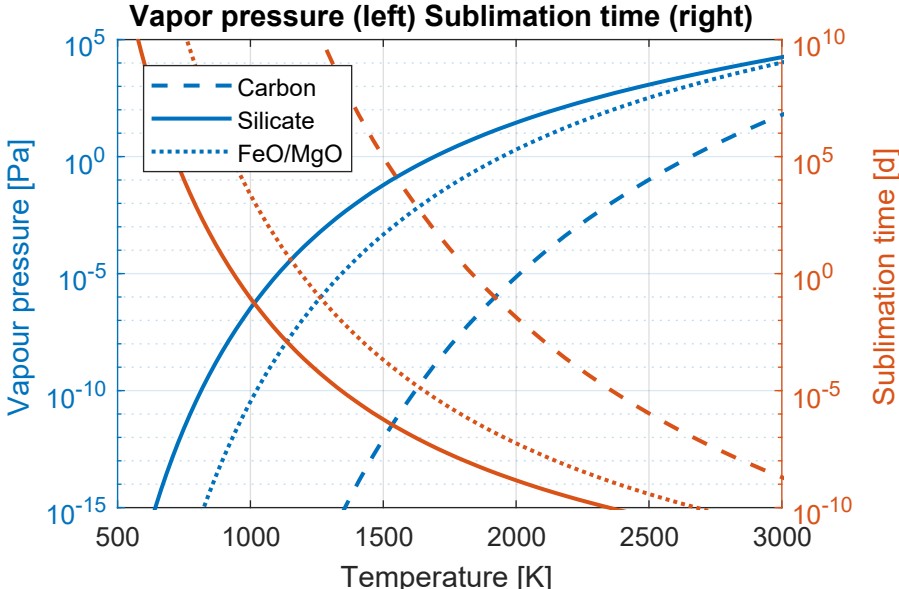

**Figure 8.** Vapour pressure (blue) and corresponding sublimation lifetime (orange) as function of 1 nm dust temperature, line styles indicate dust composition (carbon - dashed, astronomical silicate - solid, $Fe_{0.4}Mg_{0.6}O$ - dotted).





To derive the sublimation lifetime of nanometer sized dust particles, Eq. 7 is integrated using spherical geometry (Lamy, 1974):

$$t_{sub}(d, r_0) = \frac{r_0 \rho}{p_v(T_{dust}(d))} \sqrt{\frac{2\pi R T_{dust}(d)}{M}}. \tag{8}$$

Where $p_v$ is the vapour pressure of the dust material, $T_{dust}$ is the temperature of the nanodust as a function of distance from
the Sun d, and R is the universal gas constant. In Fig. 8 (orange lines and right y axis) the sublimation lifetime of 1 nm sized
dust particles made of carbon, silicate and $Fe_{0.4}Mg_{0.6}O$, is shown again within the temperature range from 500 to 3000 K. As
the vapour pressure of carbon is relatively low, the carbon nanodust has the longest sublimation lifetime. The oxides on the
other hands have much shorter lifetimes. Astronomical silicate has slightly higher vapor pressure than $Fe_{0.4}Mg_{0.6}O$ because
of its $SiO_2$ content.

As the vapour pressure is a very steep function of temperature, according to Eq. 8 the relationship is inversely translated to
the sublimation lifetime. At temperatures below 1000 K the lifetime of all different kinds of 1 nm dust are greater than $10^5$ d.
Nanodust with temperatures above 2500 K have already sublimation lifetimes below $10^{-5}$ d, these lifetimes are so short that
the dust can be regarded as non-existing.

The next step will be the direct comparison of sublimation and sputtering lifetime for nanodust within the near Sun environ-
ment.

## 4   Implications for nanodust near the Sun

In the earlier sections, it has been shown that sputtering and sublimation can be significant sinks for nanodust. The loss of
nanodust due to solar wind sputtering increases with ion number density and ion mass (see 2). The effect of sublimation
however, is a steep function of dust temperature (see Sect.3). For the comparison of sputtering and sublimation of nanodust we
have chosen the CME scenario. We find the shortest sputtering lifetimes for CME conditions, but the short duration of single
CMEs has to be taken into account.

The comparison of the lifetimes is done in the small size limit of the dust population, i.e. the sizes 0.2 nm, 1 nm, and 5 nm.
There is no experimental prove for the existence of sub nanometer dust. However, it will be hypothesized that these clusters of
molecules exist. This assumption will help to better assess the importance of nanodust sputtering in this study.

Here, we compare the sputtering and sublimation lifetimes of the three different nanodust compositions, namely carbon,
$Fe_{0.4}Mg_{0.6}O$ and silicate nanodust. Figure 9 a) shows the sputtering and sublimation lifetimes of carbon dust. All lifetimes are
compared to a duration of 2 days, which is used as the upper limit for the duration of a CME. In the case of carbon, which is a
rather sturdy material, the nanodust can survive in the near proximity of the Sun. The sublimation of carbon nanodust within
2 days occurs at a distance of 0.03 AU from the Sun, that is because of carbon's comparably high evaporation temperature of
2600K at low pressures (Whittaker, 1978). However, the sputtering lifetime of carbon is longer than sublimation counterpart.
The nanodust could withstand the sputtering of a CME to even shorter distances if it was not evaporated beforehand. When





considering the duration of a CME the sputtering and sublimation of only the smallest nanodust are comparably. In the case of carbon nanodust we state that during a typical CME sputtering is not a relevant destruction process within the inner heliosphere.

The lifetimes of $Fe_{0.4}Mg_{0.6}O$ dust for destruction by sublimation and sputtering are much shorter, see Fig. 9 b). Due to its higher vapor pressure and temperature, the $Fe_{0.4}Mg_{0.6}O$ dust sublimates already at much greater distances from the Sun compared to the carbon dust. Just below 0.2 AU all $Fe_{0.4}Mg_{0.6}O$ nanodust is evaporated within the two day period. Despite the fact that $Fe_{0.4}Mg_{0.6}O$ material is much more vulnerable to sputtering (cf. Fig. 2) as carbon, a single CME cannot destroy nanodust by a single hit. A 1 nm $Fe_{0.4}Mg_{0.6}O$ dust grain would be completely sputtered by a CME if it reached 0.1 AU but

will sublimate earlier due to its high temperature.

Regarding the sputtering and sublimation lifetime of silicate nanodust we find a different situation compared to the afore-mentioned compositions. The actual lifetimes of silicate nanodust are shown in Fig. 9 c). The sublimation lifetime of Silicate nanodust equals the two day period at distances from the Sun of around 0.15 AU. The complete sputtering of the silicate nan-odust during a CME impact occurs at solar distances of 0.2 AU, 0.07 AU and 0.03 AU for the respective grain sizes 1 nm, 5 nm

and 20 nm. We can conclude here that a region void of silicate nanodust forms after the passage of a single CME. This region lies between 0.1 and 0.15 AU for the 1-3 nm dust, larger dust rather sublimates than being fully sputtered by a single CME. The existence of this sputtering region is due to the comparably low temperatures of silicate dust that leads to lower sublimation rates for the same distances as compared to the $Fe_{0.4}Mg_{0.6}O$ dust that is destroyed by sublimation.

## 4.1   Discussion

The results shown in Fig. 9 (a-c) indicate a diverse influence of sublimation and sputtering on the nanodust environment near the Sun. The following remarks shall put the results into a context for current and upcoming dust measurements near the Sun.

When dust particles approach the Sun, they heat up quickly and along with that the sublimation becomes the governing destruction process. One finding is that sublimation for nanodust is much less size dependent compared to the sputtering process. The derived sublimation lifetimes show that the governing parameters are the distance from the Sun, the resulting

equilibrium temperature and their composition.

The sputtering process on the other hand is much more size dependent but also show distinct dependencies for dust com-position and the increasingly harsh plasma environment near the Sun. Also the type of plasma environment, i.e. slow or fast solar wind or CME impacts, present in the heliosphere play an important role for the sputtering of nanodust. The importance of sputtering for the destruction of nanodust even at 1 AU can be seen in Fig. 5 where the dust sputtering lifetimes are well below

the collisional and Poynting-Robertson lifetimes given by Grün et al. (1985). The change of the nanodust population through sputtering can result in different dust fluxes at 1 AU as expected so far. Additional measurements and dust flux modelling are needed to verify this finding.

As the sputtering and sublimation of nanodust is an important destruction process near the Sun, it can also be expected that the nanodust population changes its composition when approaching the Sun. While nanodust particles have probably a diverse

composition at 1 AU and further away, only the most durable nanodust can survive near the Sun.

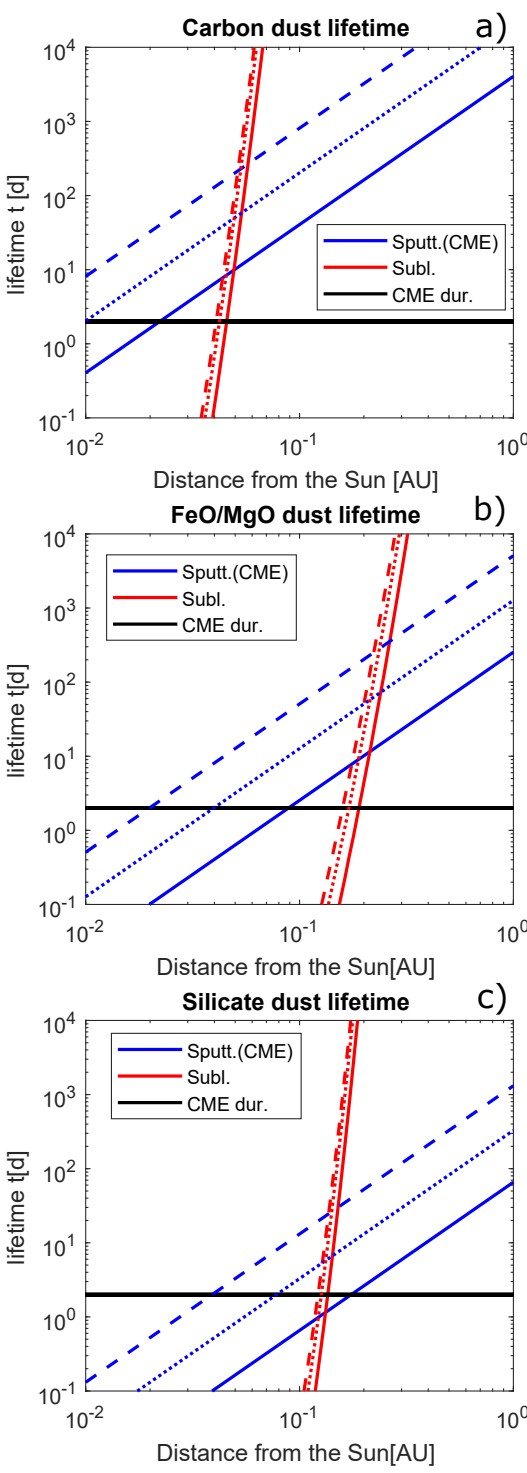

**Figure 9.** Comparison of sputtering and sublimation lifetimes of nanometersized dust near the Sun, blue lines indicate sputtering lifetimes, red lines indicate sublimation lifetimes, line style indicate dust size (dotted - 5 nm, solid - 1 nm, dashed 0.2 nm), the two day duration of a typical CME in black.





Closer to the Sun, the nanodust population becomes even more variable under the influence of CME impacts. The sputtering lifetimes of nanodust under CME conditions are several orders of magnitude lower than for the solar wind conditions (Fig. 6). Void zones for silicate nanodust in the small size limit are identified after the passage of a mature CME impact. This finding would impact the nanodust population locally and during certain times, especially at solar maximum conditions when CMEs are frequent (up to 400 per month (Lamy et al., 2019)). This variability of the nanodust population might be quantified by impact measurements onboard of Parker Solar Probe and Solar Orbiter taking sputtering and also sublimation into account. Together with the onboard plasma and optical instruments further constraints on the near Sun nanodust population are possibly deducted.

The F-corona brightness at mid infrared to visible wavelength can be attributed to thermal emission from micron sized dust particles (Kimura and Mann, 1998). Recent WISPR observations on PSP (Howard et al., 2019) show that F-corona intensity leaves its linearity around 17 solar radii (0.08 AU). These observations would support the existence of the predicted dust free zones within the F-corona (Lamy, 1974; Mann, 1992). In section 2.2 we have identified sputtering by CME impacts as a possible destruction process also for $\mu$m-dust. From Fig. 6, we find that a ten $\mu$m dust particle can be fully destroyed within three years at a distance of 0.1 AU from the Sun when struck by multiple CMEs (assuming around 100 CMEs per month) and under constant exposure to the solar wind.

In the end, it has to be noted that the given lifetimes are only valid for dust on near circular orbits. Dust affected by sublimation or sputtering is subject to a constant reduction of its size, which will result in alteration of its present orbit. The given results only represent a general description of these destruction processes. However, conclusions on the impact of sputtering and sublimation on individual dust grains along their orbits cannot be drawn and are not subject of this work.

## 5 Conclusions

Interplanetary dust enters a harsh environment when approaching the proximity of our central star. Especially the fragile nanodust is prone to destruction through sputtering by the solar wind or sublimation near the Sun. Studies on dust destructions mechanisms near the Sun already showed that there are distinct regions dominated by sublimation and sputtering in the heliosphere (e.g. Mukai and Schwehm, 1981). This study has investigated dust sputtering during more extreme conditions of Coronal Mass ejection (CME) events. CME plasma in addition to its high number density contains a large fraction of heavy ions. We find that dust is sputtered most effectively in the CME case followed by sputtering within the slow solar wind. The weakest sputtering we find in the low-density plasma of the fast solar wind. However, the sputtering process is also very composition dependent. Carbon has been found to be more stable against sputtering than the silicate and $Fe_{0.4}Mg_{0.6}O$ composition. The case of nanodust has been studied in more detail for sputtering and sublimation during a the passage of a single CME. Nanodust free zones can occur after two day CMEs for silicate (0.1 to 0.15 AU) but not for $Fe_{0.4}Mg_{0.6}O$ and carbon.

The dust component near the Sun is in the process of being probed in unprecedented detail. While Parker Solar Probe is closing in to a proximity of the Sun as close as nine solar radii, Solar Orbiter will reach 0.3 AU but also observe the Sun outside the ecliptic plane. Both missions carry instruments to measure the local electric field (Bale et al., 2016; Maksimovic





et al., 2007) which also enable the detection of dust impacts on the spacecraft. Taking into account sublimation and sputtering

will be crucial to the modelling of the measured dust fluxes. This present work gives the needed insights and want to encourage to mind these processes when interpreting the satellite measurements.

The implementation of sputtering and sublimation as destruction mechanisms needs to be included into dust flux models especially for the case of dust in the small size limit. Taking these processes into account is definitely important when considering the dust population near the Sun or other central stars. But also when considering dust trajectory modelling, the rough

environment near stars lead to a shrinking of dust particles due to sublimation and sputtering. That leads to an increase of the often used charge to mass ratio of dust in these trajectory models for the small dust component. We expect that integrating the change of the dust size together with its full equation of motion will lead to new insights of the nanodust population near central stars. A recent study by Shestakova and Demchenko (2018) derived the orbital evolution of $\mu$m dust within the sublimation zone and included the dust size reduction due to sublimation. They find either elongated dust trajectories after partial sublima-

tion or trajectories leading to complete sublimation after spiralling further into the evaporation zone of the Sun. A future study which also takes the sputtering of dust into account will find deeper insight into the fate of nanodust near the Sun and during the passage of a CME.

Variations in the F-corona intensity has usually been explained by the destruction of dust through sublimation or orbital changes (Lamy, 1974; Mann, 1992). The results of our work have shown that sputtering of micron sized dust during the

passage of multiple CME can play a role in the explanation of dust free zones in the F-corona.

Furthermore, we also expect that standard solar wind conditions can lead to significant sputtering in timescales which are shorter than the dynamical removal times of dust within intermediate distances from the Sun, i.e. 1 AU and greater.

Nevertheless, further laboratory as well as theoretical research is necessary to pin point sputtering yields for small dust grains of various composition. At the moment, experimental, theoretical and modelling results of sputtering yields show a

diverse picture where scientific consensus is missing.

*Code and data availability.* The derived sputtering yields, dust temperatures and vapor pressures are made available within the supplemented material. The MAGMA code can be obtained from Bruce Fegley upon request.

*Author contributions.* C. Baumann carried out the calculations and wrote the initial manuscript. M. Myrvang contributed the dust temperatures near the Sun. C. Baumann and I. Mann conceived the idea of the work. All authors contributed to the finalization of the manuscript.

*Competing interests.* The authors declare no competing interest.





*Acknowledgements.* This work was supported by the *Norwegian Research Council* project number 262941. We also thank the International Space Science Institute in Bern (ISSI Bern) for their support through funding of collaborative meetings of the working group 'Physics of Dust Impacts: Detection of Cosmic Dust by Spacecraft and its Influence on the Plasma Environment'. The authors thank Jan Fredrik Aasmundtveit for his contribution to the derivation of the sputtering yields.



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
