# Peer review of "Dust sputtering within the inner heliosphere: a modeling study"

_Annales Geophysicae, 2020_

## Referee Comment (RC1) · Anonymous Referee #1 · 19 May 2020

This nice paper presents modelling calculations on sputtering on dust particles, and how solar wind and CME influence the lifetime of the particles. Such studies are of importance to better understand the complex content of the F-corona, in particular in the frame of the Parker Solar Probe mission. The paper is well written, the modelling calculations are well conducted and provide new insights of the dust evolution close to the Sun. The paper can be published if the following comments are considered. Since this work is based on modelling calculations, the word "modelling" must be added in the title and in the first sentence of the abstract. Line 49: I am not sure that this comment on nanometer-sized dust particles around other stars in necessary in the paper. Line 118-125: What could be the effect on the results of the "additional parameters"? Are the results still valid even if these parameters are not considered? Perhaps the authors can provide an estimate of the uncertainties on the results presented here when not considering these parameters. Line 147: "ten orbital period"; it is orbital period of the

dust? can you give the value in day or years as for the FeMgO nanodust?

---

## Referee Comment (RC2) · Anonymous Referee #2 · 30 May 2020

Dust sputtering within the inner heliosphere

Carsten Baumann, Margaretha Myrvang, and Ingrid Mann

General Comments ——————- The paper is nicely written and addresses important aspects of small dust entering the inner solar system. The paper is recommended for publication, only minor details shall be addressed.

Detailed Comments: —————— Abstract: Instead of "particles" use "ions" since you known what your particles are.

Line 21: Spell out "WISPR", or is it "WISPER"

Line 42: Write "... as solar wind ions hit dust particles."

Line 47: Typo "exampless"

[Figure]

Line 63: Write "... bombardment by energetic ions ..."

Line 65: "... and form high quality surfaces." Perhaps you want to add the ref. J. E. Greene, Jou. Vac. Sc. Techn. A 35, 2017 , 05C204

Line 80: ".. used for the solar wind / CME conditions .." spell out "/"

Table 1: Perhaps you want to add two columns to give the (fractional ) abundance of He and heavies.

Line 91 ff: use italics for the formula symbols, same as in the formula.

Line 95: Write "... correspond to the atomic abundance ratios of the dust composition and the ion abundance ratios ...."

Figure 2: The absolute sputter yield is somewhat misleading, since only the sputter yield prorated to the solar wind ion abundance applies. Perhaps you plot the prorated sputter yield for all the solar wind ions, and sum curve that adds the yield contributions from H to Fe stepwise, thus showing their contributions. If you do that also for fast SW and CME you will get a very strong plot.

Line 100: I guess you mean "(H - Iron, Fe),"

Line 151: Write "... have sputtering lifetimes that can reach ..."

Line 179: Typo: "... assuming the dust is in at a distance ..."

Line 194 ff: use italics for the formula symbols, same as in the formula.

Line 202ff: "re-emitted" wouldn't "emitted" just do it? It is a photon of different energy anyway.

Line 209: Typo: "... material(not shown)."

Line 230: use italics for the formula symbols, same as in the formula.

Line 237: Write "... are larger than 10ˆ5 d." Remember the difference of drinking a

great cup of tea, and drinking a large cup of tea.

Line 243: Write "(see Fig. 2)." I guess.

Line 248: "prove" –> "evidence", I guess.

Line 261: Write "... at much larger distances ..."

Line 288ff: These arguments imply that the average composition of small grains changes when getting closer to the Sun. Maybe you want to elaborate on this argument.

Line 311: "the fragile nanodust" this is the first time this classification is presented. Does that imply that the nano-dust is not a solid, but small, grain, but more a composition of many grains loosley attached to each other?

Line 324ff: Sublimation seems to be important around 0.1 AU. Shouldn't there be an optical singal if you would look at optical lines of sublimated material located at about 0.1 AU. Can you address this with Solar Orbiter or PSP?

Line 341: Write "... distances from the Sun, i.e. 1 AU and further out."
* * *

---

## Short Comment (SC1) · 18 Jun 2020

This comment is to report a typo within the supplemented material.

The supplemented material uploaded contains a typo within the sputtering yield the H ion for the silicate dust material (within $supplement\_material.pdf$ and $AstronomicalSilicate.txt$):

Old version:

H,0.0059,0.0034,0.0030,0.0209,0.0109,0.0047,0.0040,0.0346,0.0103,0.004 9,**0.0391**,0.0305

Correct version:

H,0.0059,0.0034,0.0030,0.0209,0.0109,0.0047,0.0040,0.0346,0.0103,0.004 9,**0.0039**,0.0305

The corrected value does neither change the results of the manuscript nor the findings.

The figures of the manuscript do not change when using the corrected value. The original figures can still be used and the authors declare that no other changes has been made to manuscript. It was only barely visible during the production of the new Figure02 (requested from reviewer 2). The new Figure 2 has been made using the correct value.

To correct this typo, a new version of the supplemented material has been prepared which will be uploaded with the revised version of the manuscript.

---

## Author Response (AR1)

**Dust sputtering within the inner heliosphere: a modeling study**

C. Baumann, M. Myrvang, I. Mann

Response to Editor

The authors would like to thank the editor Johan De Keyser for taking care of the editorial handling of our manuscript. The very timely acquisition of two reviewers is very much appreciated.

For a better overview, our responses to both reviewers are presented on the following pages again. In addition, we have prepared a revised version of the manuscript and have compiled a document, that shows the tracked changes with respect to the initial submission. The track changes document is attached below the reviewer comments. We hope that our revision satisfies the reviewer comments and also the editor's point of view.

A technical note: The title has been changed in the revised manuscript in accordance to reviewer #1. However, the authors cannot change the title within the Annales Geophysicae publication system anymore.

Response to Reviewer #1

We very much appreciate the constructive comments on our manuscript and the overall positive judgement of our work. For the revised version all comments have been taken into account and have helped to improve the quality of our manuscript.

In the following we will address all comments point by point.

*...Since this work is based on modelling calculations, the word "modelling" must be added in the title and in the first sentence of the abstract....*

In accordance with the reviewer we have added the term 'modeling' to the title and abstract. The title has been adjusted to: 'Dust sputtering within the inner heliosphere: a modeling study'. The first sentence of the abstract has been changed to:'The aim of this study is to investigate through modeling how sputtering ...'

*...Line 49: I am not sure that this comment on nanometer-sized dust particles around other stars is necessary in the paper...*

The reviewer is correct and we have removed this sentence from the manuscript.

*...Line 118-125: What could be the effect on the results of the "additional parameters"? Are the results still valid even if these parameters are not*

*considered? Perhaps the authors can provide an estimate of the uncertainties on the results presented here when not considering these parameters....*

The reviewer points on the not well known nature of the sputtering yield for dust grains. We have added a statement how the additional but rather unknown parameters within the sputtering yield may affect our results. 'Due to a lack of quantitative information on these enhancements factors for our study we use the conservative sputtering yields given by SRIM. As a consequence, our results provide an upper limit for dust sputtering lifetimes. We speculate that dust sputtering lifetimes could be one order of magnitude shorter when taking the microphysics of dust sputtering into account.'

*...Line 147: "ten orbital period"; it is orbital period of the dust? can you give the value in day or years as for the FeMgO nanodust?...*

The reviewer is right and we have changed this cumbersome expression to 'ten years'.

Response to Reviewer #2

We would like to thank the reviewer for his or her substantial review of our manuscript. The overall positive appraisal of our work leaves us feeling grateful. In addition, the minor corrections to manuscript and especially the suggestions to the content by the reviewer are very much appreciated. All comments have found its way into the manuscript and will be addressed point by point below.

*...Abstract: Instead of "particles" use "ions" since you known what your particles are. ....*

*...Line 21: Spell out "WISPR", or is it "WISPER" ....*

*...Line 42: Write "... as solar wind ions hit dust particles."...*

*...Line 47: Typo "exampless"...*

*...Line 63: Write "... bombardment by energetic ions ......*

The minor corrections by the reviewer have been applied to the mansucript.

*...Line 65: "... and form high quality surfaces." Perhaps you want to add the ref. J. E. Greene, Jou. Vac. Sc. Techn. A 35, 2017 , 05C204...*

We give thanks to the reviewer for this suggestion. This publication is indeed a very valuable reference and fits perfectly. We have added it to the manuscript.

*...Line 80: ".. used for the solar wind / CME conditions .." spell out "/"...*

We have changed the manuscript accordingly.

*...Table 1: Perhaps you want to add two columns to give the (fractional ) abundance of He and heavies. ....*

We thank the reviewer for the suggestion to add the abundances to the table and have done so.

Table 1: Solar wind properties as used within this study. Helium and heavy ion fraction of the solar wind is used from Killen2012.

|  | plasma density $n_p$ | speed $v_p$ | He fraction | Heavy ion frac. |
| --- | --- | --- | --- | --- |
| slow SW | $8\,\mathrm{cm}^{-3}$ | $300\,\mathrm{km/s}$ | 4% | 0.11% |
| fast SW | $3\,\mathrm{cm}^{-3}$ | $800\,\mathrm{km/s}$ | 2% | 0.06% |
| CME | $70\,\mathrm{cm}^{-3}$ | $500\,\mathrm{km/s}$ | 30% | 3% |

*...Line 91 ff: use italics for the formula symbols, same as in the formula.*

*...Line 95: Write "... correspond to the atomic abundance ratios of the dust composition and the ion abundance ratios .......*

We have changed the manuscript in accordance with the corrections by the reviewer.

*...Figure 2: The absolute sputter yield is somewhat misleading, since only the sputter yield prorated to the solar wind ion abundance applies. Perhaps you plot the prorated sputter yield for all the solar wind ions, and sum curve that adds the yield contributions from H to Fe stepwise, thus showing their contributions. If you do that also for fast SW and CME you will get a very strong plot. ....*

The reviewer is definitely correct about the absolute sputtering yield. We have replaced the old Figure 2 with a new Figure that shows in a stacked bar plot the solar wind composition prorated sputtering yield. We have also added a description of this new

[Figure]

Figure 1: Solar wind ion prorated sputtering yield for $Fe_{0.4}Mg_{0.6}O$, silicate and carbon. Sputtering yields are a function of solar wind ion itself, its fractional abundance (fast solar wind (fSW), slow solar wind (sSW) and CME), and its mean speed.

figure. However, we still mention the absolute sputtering yield as it is available in the supplemented material.

The description of Figure 2 has been changed to: 'Figure 2 shows the results of the assessment, i.e. the sputtering yield as a function of ion species (H - S), for the three different materials. The given values are not absolute but prorated with solar wind ion composition present in fast and slow solar wind as well as CME conditions ($Y_{i,k} \cdot c_k$, $c_k$ is the fractional abundance of ion k in the solar wind conditions, cf. Eq. 1, Tab. 1). The highest sputtering yields are found for $Fe_{0.4}Mg_{0.6}O$ material, the yields are somewhat smaller for silicate and are the lowest by far for carbon material. The Figure 2 also shows that the sputtering yields strongly increase during CME conditions and that this is due to the sputtering by the heavy ions that are more abundant during CME than in the normal solar wind. Likewise, the higher abundance of He-ions in the slow solar wind explains why sputtering yields are larger in the slow solar wind then in the fast solar wind.

*...Line 100: I guess you mean "(H - Iron, Fe),"...*

*... Line 151: Write "... have sputtering lifetimes that can reach ..."...*

*...Line 179: Typo: "... assuming the dust is in at a distance ..."*

*...Line 194 ff: use italics for the formula symbols, same as in the formula. ...*

*...Line 202ff: "re-emitted" wouldn't "emitted" just do it? It is a photon of different energy anyway. ...*

*...Line 209: Typo: "... material(not shown)." ...*

*...Line 230: use italics for the formula symbols, same as in the formula.*

*...Line 237: Write "... are larger than 10^5 d." Remember the difference of drinking a great cup of tea, and drinking a large cup of tea.*

*...Line 243: Write "(see Fig. 2)." I guess.*

*...Line 248: "prove" −¿ "evidence", I guess.*

*...Line 261: Write "... at much larger distances ...*

We would like to thank the reviewer for identifying the typos and small mistakes in the text. We have corrected all the comments.

*...Line 288ff: These arguments imply that the average composition of small grains changes when getting closer to the Sun. Maybe you want to elaborate on this argument. ....*

The reviewer raises a valid point on the dust composition changes in the proximity of the Sun. We have added a short paragraph, which shortly discusses this point based on our model calculations. However, quantitative statements are not possible from our results.

'Our calculation allow the assumption that the majority of nanodust in the close proximity of the Sun is made of carbon. $Fe_{0.4}Mg_{0.6}O$ and silicate dust is very likely sublimated or sputtered and not very abundant there. Quantitative statements on the abundance of different dust species depends also on their production rates near the Sun. Giving production rates for dust and nanodust made of different material are beyond the scope of this study.'

*...Line 311: "the fragile nanodust" this is the first time this classification is presented. Does that imply that the nano-dust is not a solid, but small, grain, but more a composition of many grains loosely attached to each other? ....*

The reviewer refers to a general point in dust science. There are different types of dust, on the one hand rock solid fragments of small solar system bodies and on the other hand fragile conglomerates of smaller parts that form a larger dust grain. In the case of nanodust, its structure can be assumed as clusters of atoms and molecules. In the case of atomic clusters the binding energy might be as larger as in solid grains. Molecular clusters are much weaker bound and the term 'fragile' might be used for these clusters.

Our intention using the term 'fragile' related to the short lifetimes of nanodust, so that a single CME might be enough to destroy a nanometersized dust grain. In addition, we did not aim to relate to the idea mentioned by the reviewer as our calculation does not cover this perspective of the dust and nanodust. For clarification, we have changed

the term 'fragile' to 'small' within the manuscript.

> *...Line 324ff: Sublimation seems to be important around 0.1 AU. Shouldn't there be an optical singal if you would look at optical lines of sublimated material located at about 0.1 AU. Can you address this with Solar Orbiter or PSP? ...*

The reviewer points out a good possibility for future research. PSP carries a Wide field visible light imager (WISPR) that is not able to do spectroscopic measurements. Solar orbiter on the other hand carries different spectrometers for the EUV- and X-rays. We have added the following paragraph to account for that idea.

'An additional possibility to characterize the composition of dust near the Sun is the detection of emission lines from sublimated dust atoms or ions. At 0.1 AU sublimation starts to be effective and might lead to layers of atomic species. Also collisional dust destruction can be a source of ions which might be visible near the sun (Mann et al. 2005). These ions might be detected optically from specific emission lines or using in-situ mass spectrometric measurements onboard spacecraft.'

> *...Line 341: Write "... distances from the Sun, i.e. 1 AU and further out....*

We have implemented the correction.

**Standard comment: Typo within the supplemented material**

This comment is to report a typo within the supplemented material.The supplemented material uploaded contains a typo within the sputtering yieldthe H ion for the silicate dust material (within *supplement_material.pdf* and *AstronomicalSilicate.txt*):

Old version: H,0.0059,0.0034,0.0030,0.0209,0.0109,0.0047,0.0040,0.0346,0.0103,0.0049,**0.0391**,0.0305

Correct version: H,0.0059,0.0034,0.0030,0.0209,0.0109,0.0047,0.0040,0.0346,0.0103,0.0049,**0.0039**,0.0305

The corrected value does neither change the results of the manuscript nor the findings.The figures of the manuscript do not change when using the corrected value. The original figures can still be used and the authors declare that no other changes has been made to manuscript. It was only barely visible during the production of the new Figure 2 (requested from reviewer 2). The new Figure 2 has been made using the correct value. To correct this typo, a new version of the supplemented material has been prepared which will be uploaded with the revised version of the manuscript.

[revised manuscript text omitted]